

# The flow past a flatback airfoil with flow control devices: Benchmarking numerical simulations against wind tunnel data

George Papadakis[1], Marinos Manolesos[2]

[1]School of Naval Architecture & Marine Engineering, National Technical University of Athens, Athens, Greece
[2]College of Engineering, Swansea University, Swansea, UK

*Correspondence to*: George Papadakis (papis@fluid.mech.ntua.gr)

**Abstract.** As wind turbines grow larger, the use of flatback airfoils has become standard practice for the root region of the blades. Flatback profiles provide higher lift and reduced sensitivity to soiling at significantly higher drag values. A number of

flow control devices has been proposed to improve the performance of flatback profiles. In the present study, the flow past a flatback airfoil at a chord Reynolds number of $1.5 \times 10^6$ with and without trailing edge flow control devices is considered. Two different numerical approaches are applied, Unsteady Reynolds Averaged Navier Stokes (RANS) simulations and Detached Eddy Simulations (DES). The computational predictions are compared against wind tunnel measurements to assess the suitability of each method. The effect of each flow control device on the flow is examined based on the DES results on the

finer mesh. Results agree well with the experimental findings and show that a newly proposed flap device outperforms traditional solutions for flatback airfoils. In terms of numerical modelling, the more expensive DES approach is more suitable if the wake frequencies are of interest, but the simplest 2D RANS simulations can provide acceptable load predictions.

## 1 Introduction

Wind turbine blade design is dominated by structural and transportation requirements in the root region, which results in

compromised aerodynamic design. This leads to the use of thicker airfoils with a blunt Trailing Edge (TE), i.e. flatback (FB) airfoils, in the inner part of the blade. This design option becomes more and more popular as wind turbines grow larger and rotor diameters go beyond 200m.

FB airfoils provide a number of aerodynamic, structural and aeroelastic benefits compared to traditional airfoils of the same

thickness. Aerodynamically, they provide higher lift values due to the reduced adverse pressure gradient over the aft part of the suction side. Additionally, their performance is insensitive to surface roughness compared to traditional airfoils of similar thickness (Baker et al., 2006). Structurally, they have larger cross-sectional area and can lead to significant blade weight reduction (Griffith and Richards, 2014). Blades that utilize FB profiles also have improved aeroelastic behaviour, as the blunt TE offers increased flapwise stiffness.




The performance of FB profiles can be further improved by means of various TE add-ons, which alter the unsteady bluff-body wake that forms downstream of the blunt TE. TE devices extend the vortex formation length at the wake of the blunt TE (Manolesos et al., 2016) and lead to increased base pressure and Strouhal number of the flow fluctuations in the wake. At the same time the amplitude of the flow fluctuations is reduced, reducing the danger for vortex induced vibrations for wind turbine

blades and also reducing noise (Barone and Berg, 2009).

The numerical study of the flow past FB airfoils is particularly challenging due to its unsteady and three-dimensional (3D) nature, which includes impingement, separation and vortex shedding. Prediction of force coefficients and wake characteristics depend heavily on the fidelity of the numerical approach, the mesh resolution and the size of the domain (Calafell et al., 2012;

Lehmkuhl et al., 2014; Stone et al., 2009). It is generally agreed that two-dimensional and Reynolds Averaged Navier Stokes (RANS) non-physically damp the flow (Stone et al., 2006) and that higher fidelity approaches might be more suitable (Standish and Van Dam, 2003), especially if the flow frequencies in the wake are of interest. There has been a number of investigations at high Re numbers (Calafell et al., 2012; Kim et al., 2014; Lehmkuhl et al., 2014; Standish and Van Dam, 2003; Stone et al., 2009; Xu et al., 2014). However, most deal with 2D or low aspect ratio simulations and not with TE devices for flow control.

The highest fidelity simulations to date remain that of (Hossain et al., 2013), who performed DNS on a FX77-W-500 airfoil, albeit at a very low Re number of Re = 3900.

As far as bluff body drag reduction is concerned, it has been an active area of research for decades (Choi et al., 2008; Tanner, 1975; Viswanath, 1996). The most commonly applied technique to reduce base drag is the alteration of the TE shape, either

by means of afterbodies or by 3D shaping of the TE. The common aim is to increase base pressure by modifying the vortex shedding in the wake. Studies on flatback airfoils (Baker and van Dam, 2008; Kahn et al., 2008; Krentel and Nitsche, 2013; Mayda et al., 2008; Nash, 1965) have highlighted a splitter plate or a pair of them, forming a cavity at the TE, as an extremely simple and effective method.

The present investigation deals with the flow past a 30% thick wind turbine airfoil with a 10% thick blunt TE (Boorsma et al., 2015) at a Reynolds number of $Re = 1.5 \times 10^6$. The corresponding Reynolds number based on the blunt TE thickness is $Re_{TE} = 1.5 \times 10^5$. It is the continuation of an experimental survey that examined a number of TE devices to improve the performance of the profile (Manolesos and Voutsinas, 2016). The latter work was a proof-of-concept study for a FB airfoil Flap device, which proved to perform better than traditional devices.


The objective of the present study is (a) to examine which numerical approach is most suitable to study the flow in question and (b) to provide insight into the effect of the various flow control devices on the airfoil wake. The next section presents the methodology followed in this study. Subsequently, the results are presented and discussed, while the final section concludes the findings of this investigation.



## 2 Methodology

### 2.1 Numerical set-up


All simulations were performed using MaPFlow (Papadakis and Voutsinas, 2019), the CFD solver developed at NTUA's Aerodynamics Laboratory. In the present case, the Eulerian module of MaPFlow was

used, which is a compressible, cell centred CFD Solver that can use both structured and unstructured grids. Turbulence closures implemented on MaPFlow include the one-equation Spalart Allmaras (SA) turbulence model (Spalart and Allmaras, 1992) well as the two-equation turbulence model of Menter, k-ω SST, (Menter, 1993). The Improved Delayed

Detached Eddy Simulation (IDDES - from here on DDES) approach implemented in MaPFlow follows the suggestions of (Shur et al., 2008).

The DDES model definition for the local length scale Δ used in the current work is :

$$\Delta = min[max(C_w d_w, C_w h_{max}, h_{wn}), h_{max}] \qquad (1)$$

where $C_w$ is an empirical constant, $h_{wn}$ is the mesh step in the wall normal direction and $h_{max} = max(\Delta x, \Delta y, \Delta z)$. For $C_w$, the value of 0.15 is chosen.

Switch from RANS to DDES brand is accomplished by altering the distance from the wall used in the original model according to:

$$l_{DDES} = l_{RANS} - f_d max(0, l_{RANS} - l_{LES}) \qquad (2)$$

$l_{RANS}$ is the distance from the wall (original SA model) and $l_{LES} = C_{DES} \Psi \Delta$ where $C_{DES}$ is an empirical constant and $\Psi$ a low Reynods number correction (for details see (Shur et al., 2008))

In the present study 2D and 3D simulations using the Unsteady RANS

approach with the SA turbulence model and the DDES method were performed. For the 3D results two grids of different spatial resolution were generated. The grids were hybrid, structured close to the airfoil and

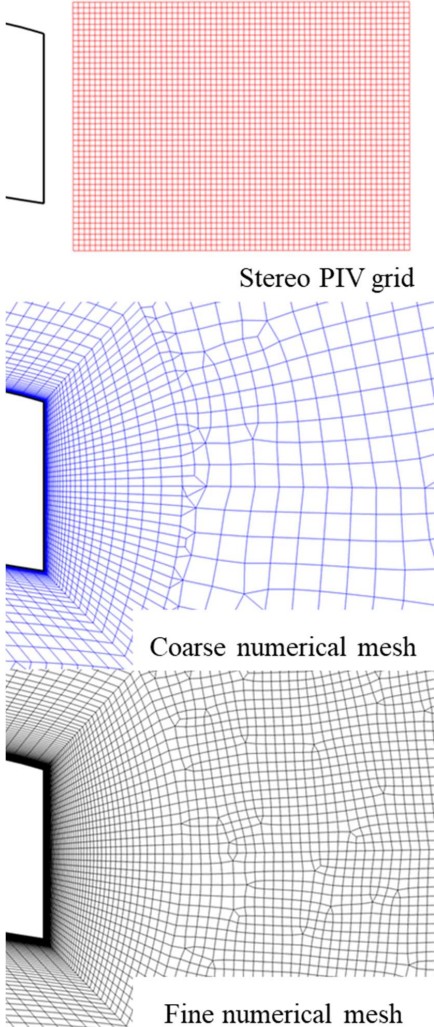

**Figure 1.** Details of the numerical and Stereo PIV grids. Top: Stereo PIV measurement grid. Middle: Coarse numerical mesh (5M cells). Bottom: Fine numerical mesh (25M Cells).



unstructured far from it. Figure 1 shows the Coarse and the Fine grids along with the grid used in the Stereo PIV measurements
for comparison. The coarser grid consisted of 5M cells, while the finer contained 25M cells. The computational domain
extended for one chord length in the spanwise direction, i.e. much wider than the spanwise coherence in the wake, which has
been found to be ~5 TE heights (Metzinger et al., 2018). The reason for this is that the present study will be extended to higher
AoA, where Stall Cells appear on flatback airfoils (Manolesos et al., 2013; Manolesos and Voutsinas, 2016, 2014). For the
accurate computation of these large structures high AR computational domains are necessary. Symmetry conditions were
applied at the side boundaries of the computational domain. For the coarser grid, 60 nodes were used in the spanwise direction
while 200 were defined for the finer grid. The grids were refined in all directions to ensure that the computational cell aspect
ratio in the wake remained close to unity. Regarding the farfield boundary, it was located 100 chords away from the wing to
minimize the influence of the external boundary conditions on the simulations (Sørensen et al., 2016). The same computational
mesh was used for both URANS and DES simulations to exclude grid related differences. For the 2D URANS simulations, a
slice of the Coarse grid was used, which had 88k cells. All simulations were performed at an angle of attack (AoA) α = 0°.


### 2.2 Experimental set-up

The numerical results are compared against Stereo Particle Image Velocimetry (PIV), force coefficient and hot wire
measurements in the wake of the airfoils. In the experiments, the wing model had a chord of c = 0.5m and an Aspect Ratio of
AR = 2. Figure 2 presents a schematic of the measurement set up, where the Stereo PIV measurement plane is shown along
with the location of the hot wire probe in the wake of the airfoil. Extensive details of the wind tunnel tests can be found in
(Manolesos and Voutsinas, 2016).

All cases concern a 30% thick FB airfoil with a 10.6% thick TE (LI30-FB10 (Boorsma et al., 2015), Figure 2), which was
tested at $Re_c$ = 1.5×10⁶. The four best performing devices tested by (Manolesos and Voutsinas, 2016) are considered in the
present investigation, as shown in Figure 3. The Splitter and the Offset Cavity are traditional devices studied by a number of
researchers, e.g. (Kahn et al., 2008; Viswanath, 1996). The Flap and the Flap + Offset Cavity where proposed in a proof of
concept study (Manolesos and Voutsinas, 2016). The TE thickness was $h_{TE} = 53mm$, while the Splitter and Offset Cavity
lengths were $0.81h_{TE}$. The Offset Cavity plates were located $0.19h_{TE}$ off the TE edges. The Flap had a chord of 0.62h and
was located at 20° with respect to the airfoil chord. Its TE was at $0.75h_{TE}$ downstream of the airfoil TE. In the remaining of
this article all quantities are non-dimensionalized using the chord and the free stream velocity, $V_\infty$, as reference values, unless
otherwise stated. A constant misalignment of the model has been aligned of the model has been allowed for in the results.





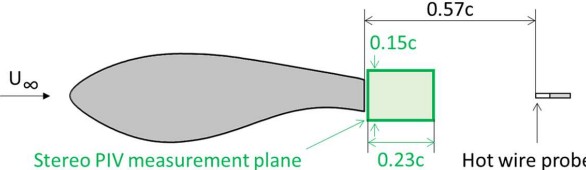

Figure 2: Schematic of the experimental set-up, showing the flatback airfoil under investigation, LI30-FB10, the location of the hot wire probe in the wake and size of the Stereo PIV measurement plane.

| Splitter | Offset Cavity | Flap | Flap + Offset Cavity |
|---|---|---|---|

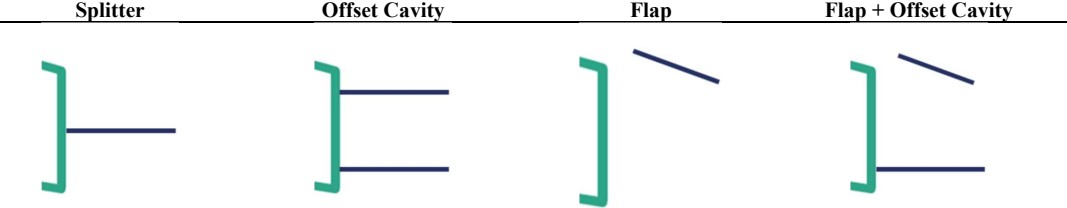

Figure 3. Close up detail of the blunt trailing edge with the four devices considered in this study.

## 3 Results and Discussion

Given the large amount of numerical and experimental data only a selection is presented in this report. First, the numerical predictions are compared against the measurement data in terms of force coefficients, wake frequencies and amplitudes. Then the time-averaged Stereo PIV data are compared against the simulation results. The Coarse and Fine URANS and DDES

results are included in these comparisons to assess which method is the most suitable for the analysis of the flow under investigation. The 2D URANS data are also included for reference. Finally, the most suitable method, DDES on the Fine mesh, is used to examine the flow mechanisms introduced by the flow control devices at the TE of the wing.

### 3.1 Comparison with experimental data

#### 3.1.1 Force coefficients

Figure 4 compares the lift and drag coefficient predictions with the relevant experimental values, while the glide ratio (L/D) comparison is given in Figure 5. The error bars are based on the standard deviation values. For completeness the relevant values are also given in Appendix

Table 2 and Table 3, in the Appendix. In terms of mean values, the agreement is good in qualitative terms and all methods capture the trends for lift, drag and Lift to Drag ratio (L/D). In quantitative terms predictions are significantly better for lift

than for drag, in agreement with previous studies (Stone et al., 2009; Xu et al., 2014). As far as comparative performance is concerned, all simulations, even the lowest fidelity 2D RANS, agree with the experiments and suggest that the airfoil with the Flap would produce the highest lift and the highest Lift to Drag ratio, see Figure 5.



With regards to load variation (denoted by the error bars for the Lift (Cl) and Drag (Cd) coefficients in Figure 4) , only the
highest fidelity method, DDES on the Fine mesh, predicts unsteady forces for all cases with or without the TE devices. There
are no experimental data for the load variation, but Stereo PIV and hot wire velocity measurements indicated significant flow
variation in the wake, so it is expected that loads on the wing should vary as well.

With regards to the different methods, URANS predictions are significantly affected by the change from 2D to 3D. 2D
simulations overpredict drag, as they restrict a naturally three-dimensional flow to be two-dimensional (Metzinger et al., 2018;
Xu et al., 2014). This leads to very high load fluctuations especially for the Plain airfoil case. As the wake analysis in the
following section indicates, the shed vorticity remains coherent and thus the induced loads are overpredicted. On the contrary,
URANS simulations, 2D or 3D, do not predict any load fluctuations for the cases with the TE devices. It appears that URANS
artificially damps the flow, regardless of the dimensions or the spatial resolution when TE devices are employed.


Mesh resolution appears to play an important role for DDES, more so than for URANS. URANS predictions remain practically
unchanged moving from the Coarse to the Fine mesh, especially for the TE device cases. DDES predicts higher drag values
with the Fine mesh, and more importantly, the Fine DDES simulation is the only one that predicts unsteady loads for all cases.
Regarding URANS, unsteady loads are only predicted for the Plain airfoil case. The 2D URANS predict significantly higher
loads, a result of confining an inherently 3D flow in two dimensions.

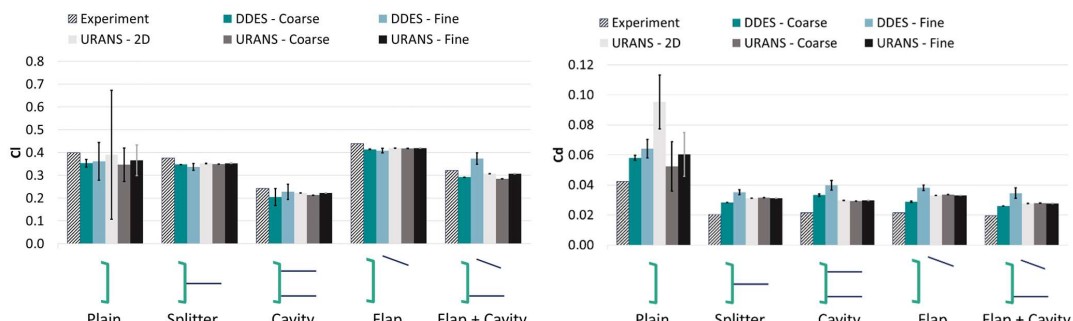

**Figure 4: Lift (left) and drag coefficient (right) values for the Plain airfoil and the airfoil with different TE devices. Comparison
between Experiments, DDES and URANS simulations. Error bars in the numerical predictions are equal to the standard deviation
of the force coefficient.**



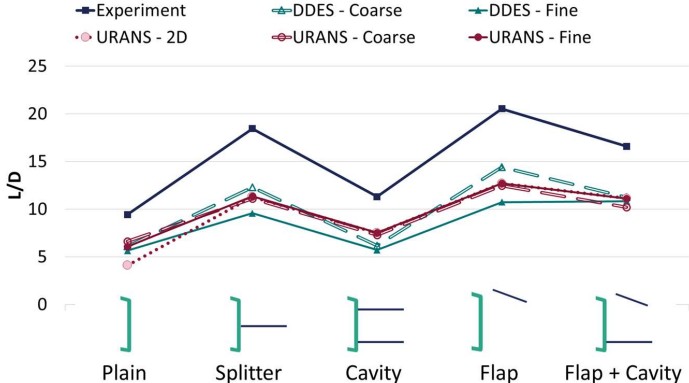

**Figure 5: Lift to Drag ratio for the Plain airfoil and the airfoil with different TE devices. Comparison between Experiments, DDES and URANS simulations.**

### 3.1.2    Wake frequencies

Figure 7, left, presents the Strouhal number values, comparing Experiments, DDES and URANS simulations. The St number is calculated based on the TE thickness and the main frequency, $f$, according to the equation below:

$$St = fh_{TE}/V_\infty \tag{3}$$

In the wind tunnel data, frequency, $f$, is the main frequency in the velocity spectrum from the wake hot wire measurements (see also Figure 2). Similarly, for all simulations the velocity timeseries at the location of the hot wire probe was examined and the dominant frequency from the vertical velocity spectrum was taken. Figure 7, right, shows the normalised peak

amplitude of that frequency, $A_{norm}$, according to Equation (4)

$$A_{norm} = \frac{A}{A_{Plain}} \tag{4}$$

where $A_{Plain}$ is the amplitude of the dominant frequency for the Plain airfoil case. Table 1 presents the relevant values ($St$ and $A_{norm}$) for the experiments and the highest fidelity simulation, Fine DDES. The Power Spectral Density (PSD) for the Fine DDES cases is given in Figure 6.

In case of the Plain airfoil, all $St$ predictions are close to the experimental results, but lower, in agreement with (Stone et al., 2009). For RANS, the wake frequency is practically insensitive to mesh resolution or dimensions (2D or 3D). Increasing the mesh size from 5M to 25M leads to improved DDES predictions for all cases. The results show that URANS fail to predict the wake unsteadiness in the case of the TE devices or provide erroneous predictions (URANS – Fine, Splitter).




A closer look at the comparison between the Experiment and the Fine DDES simulation results (Figure 7, right and Table 1)
reveals that DDES can qualitatively predict both the frequency and the normalized peak amplitude, suggesting that the main
flow mechanisms are correctly captured by the selected approach. The largest difference is observed for the Flap and Flap +
Offset Cavity cases. At this point the reason for this discrepancy remains unclear. It is noted however, that during the
experiments the Flap, which was manufactured out of a 2 mm thick aluminium sheet, was slightly deflected under the
aerodynamic load. It was not possible to quantify the deflection or its effect at the time. Interestingly, DDES predictions
regarding the velocity's fluctuation frequency are in fair agreement with the experimental data for all configurations even when
the coarser grid is employed.

According to (Manolesos and Voutsinas, 2016), the wake contains structures of smaller size and higher frequency, when the
TE devices are used. The velocity spectrum from the DDES simulations (Figure 6) agrees with this observation except for the
Flap case, where the St number is the same as for the Plain airfoil case. It is conceivable that this is because the vortex shed
from the lower edge of the blunt TE is unaffected by the presence of the Flap, as discussed in Section 3.2.

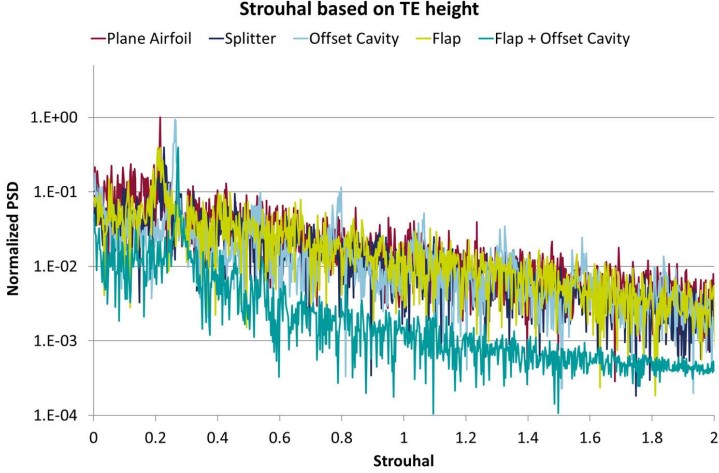

**Figure 6: Normalized Power Spectral Density of the velocity time series for the Plain airfoil and all the TE device cases from Fine
DDES simulations. The vertical velocity is considered at (x, y) = (1.57c, 0) see also Figure 2. Strouhal number is defined based on
the trailing edge thickness, $St = f h_{TE}/V_\infty$**




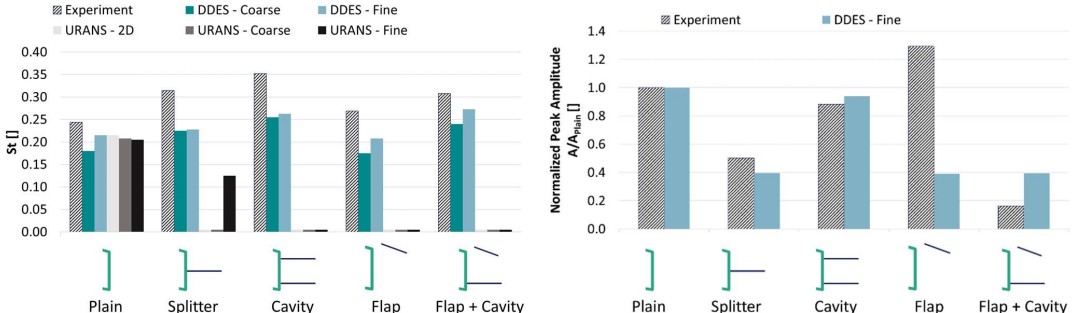

**Figure 7: Left: Strouhal number ($St = fh_{TE}/V_\infty$) values for the Plain airfoil and the airfoil with different TE devices. Comparison between Experiments, DDES and URANS simulations. Right: Normalized peak amplitude of the frequency spectrum Comparison between Experiments and Fine DDES simulations.**

|  | Strouhal number | | Normalized Peak Amplitude | |
|---|---|---|---|---|
|  | Experiment | DDES | Experiment | DDES |
| **Clean Airfoil** | 0.24 | 0.21 | 1.00 | 1 |
| **Splitter** | 0.31 | 0.23 | 0.50 | 0.40 |
| **Offset Cavity** | 0.35 | 0.26 | 0.88 | 0.94 |
| **Flap** | 0.27 | 0.21 | 1.30 | 0.39 |
| **Flap + Offset Cavity** | 0.31 | 0.27 | 0.16 | 0.39 |

**Table 1: Strouhal number ($St = fh_{TE}/V_\infty$) and normalized peak amplitude of the frequency spectrum for the Plain airfoil and the airfoil with different TE devices. Comparison between Experiments and DDES**

### 3.1.3 Wake velocity and turbulence quantities

In this section the time-averaged velocity magnitude as measured on the Stereo PIV plane is compared with the numerical predictions on the Fine mesh. In the interest of brevity only the $\overline{v'v'}/V_\infty^2$ normal Reynolds stress is presented from the

turbulence quantities, but it is noted that the same conclusions can be reached by examining the other quantities. Figures from Figure 8 to Figure 12 present the relevant contours for all the examined cases.

In the Plain airfoil case (Figure 8), where both URANS and DDES predict significant vortex shedding, the velocity contours are very similar, with the DDES prediction closer to the wind tunnel data. In all cases, the mean velocity wake becomes initially

thinner (as highlighted by the black arrows in Figure 8) before expanding further downstream. This is linked to the roll up of the two shear layers into discrete vortices which are shed in the wake. The recirculation region compares well between experiments and simulations, with numerical prediction of the recirculation length being similar to the measured one.



Contrary to mean velocity contours, the Reynolds stress predictions, differ to some extent, with the URANS predicted values
being smaller than the DDES and the experimental ones. Again, there is a very good agreement between the latter two. The
DDES velocity fluctuation peaks appear at the same locations as in the experiments, which means that that the vortex formation
length (Tombazis and Bearman, 1997; Williamson, 1996) is also predicted correctly.

In all other cases (Splitter, Cavity, Flap, Flap + Offset Cavity) the URANS simulations do not predict vortex shedding. As a
result, the recirculation region is significantly longer than in the wind tunnel tests and Reynolds stress values are close to zero
everywhere. Additionally, the lack of mixing in the wake means that low velocities are maintained in the wake further
downstream. This is not in agreement with the experimental measurements or the DDES predictions, where vortex shedding
is observed along with significant Re stress values and quicker wake recovery.

Figure 12 reveals a disagreement between the DDES Re stress predictions and the wind tunnel measurements for the Flap +
Offset Cavity case. The Stereo PIV data, in agreement with the hot wire measurements, suggest that the flow variations are
significantly smaller for this case than all other cases. DDES do predict reduced fluctuations compared to the Plain airfoil case,
but not to the extent suggested by the measurements. This is also confirmed by the PSD graph shown in Figure 6, where the
energy contained in all frequencies of the Flap + Offset Cavity case are lower than all other cases. However, the recirculation
region and vortex formation length remain larger in the DDES simulations than in the wind tunnel tests.

In general, the DDES predictions are satisfactory but the wake appears to be 'wider', or more diffused, than in the experiments.
This could imply that numerical diffusion is significant even with the 25M cells mesh and that possible improvements can be
expected by further increase of the spatial resolution.




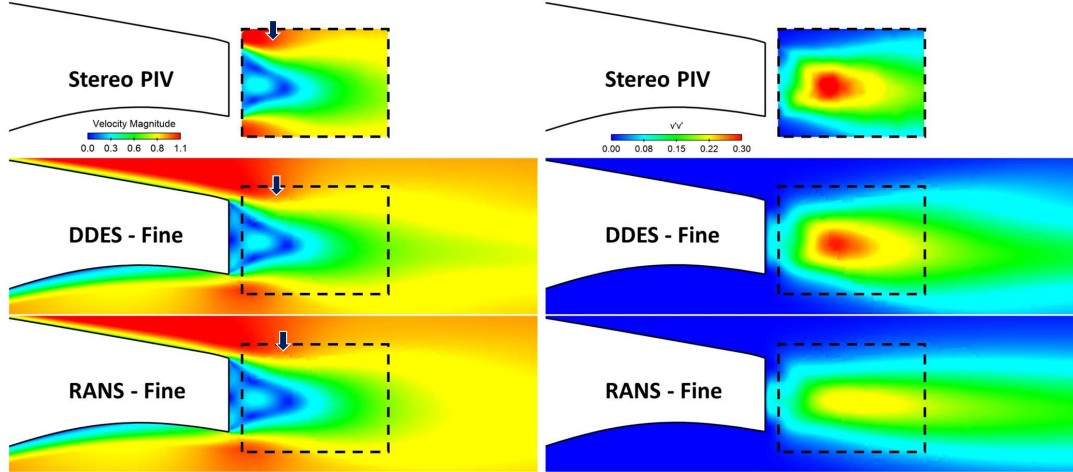

**Figure 8:** Plain airfoil case. Normalized velocity magnitude contours, $V/V_\infty$, (left) and $\overline{v'v'}/V_\infty^2$ normal Reynolds stress contours (right). Comparison between Stereo PIV, RANS and DDES results. The Fine mesh (25M cells) was used in the calculations.

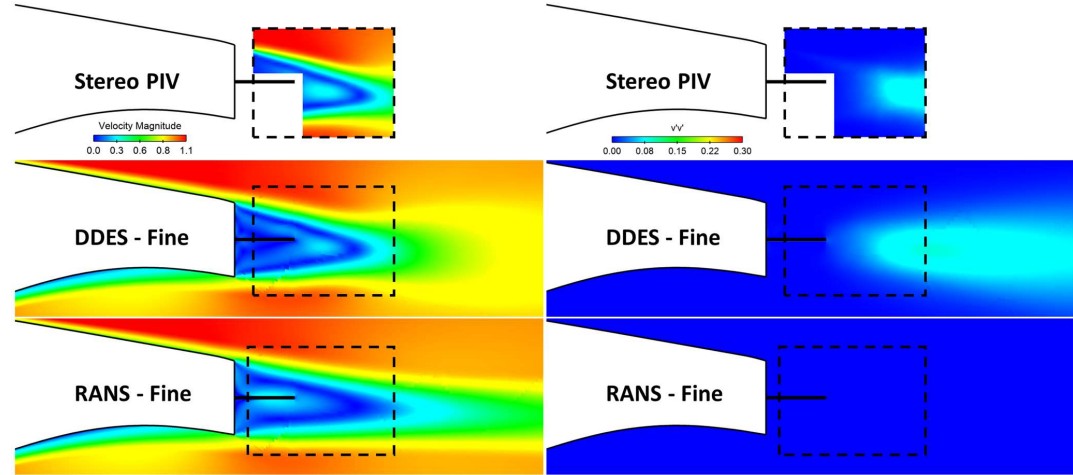


**Figure 9:** Splitter case. Normalized velocity magnitude contours, $V/V_\infty$, (left) and $\overline{v'v'}/V_\infty^2$ normal Reynolds stress contours (right). Comparison between Stereo PIV, RANS and DDES results. The Fine mesh (25M cells) was used in the calculations.





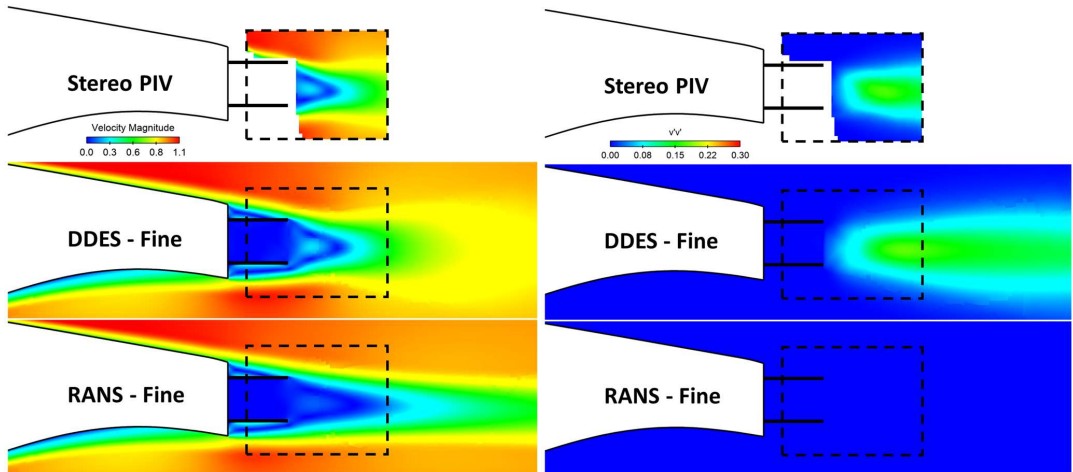

**Figure 10: Offset Cavity case. Normalized velocity magnitude contours, $V/V_\infty$, (left) and $\overline{v'v'}/V_\infty^2$ normal Reynolds stress contours**
**(right). Comparison between Stereo PIV, RANS and DDES results. The Fine mesh (25M cells) was used in the calculations.**

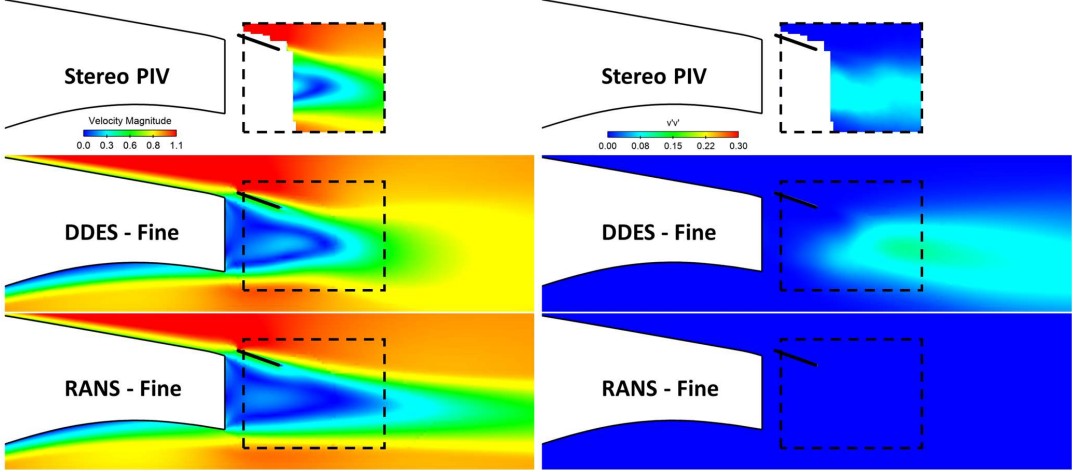

**Figure 11: Flap case. Normalized velocity magnitude contours, $V/V_\infty$, (left) and $\overline{v'v'}/V_\infty^2$ normal Reynolds stress contours (right). Comparison between Stereo PIV, RANS and DDES results. The Fine mesh (25M cells) was used in the calculations.**




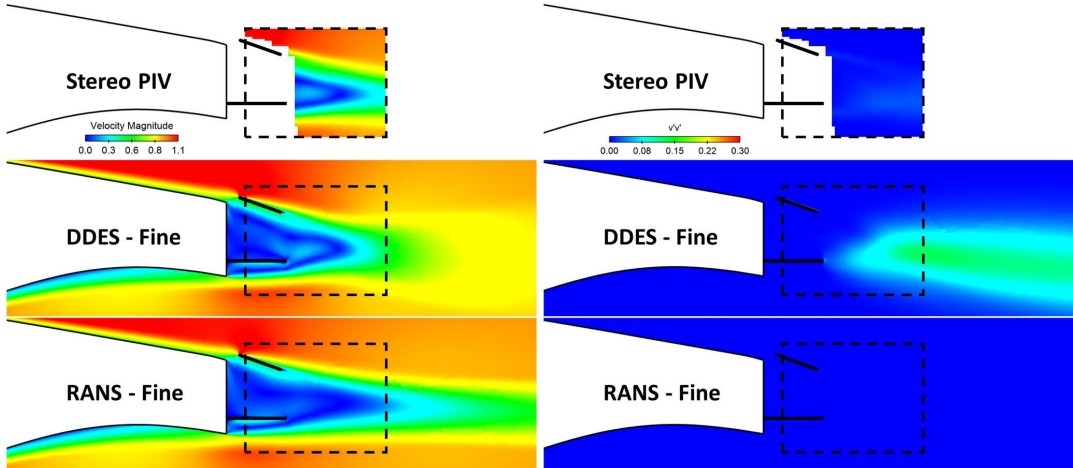

**Figure 12: Flap + Offset Cavity case. Normalized velocity magnitude contours, $V/V_\infty$, (left) and $\overline{v'v'}/V_\infty^2$ normal Reynolds stress contours (right). Comparison between Stereo PIV, RANS and DDES results. The Fine mesh (25M cells) was used in the calculations.**

### 3.2 Wake development

The analysis so far has shown that the predictions of the highest fidelity numerical approach, DDES on the Fine mesh, are closer to the experimental data in terms of load values, wake velocities and frequencies and turbulence quantities. Consequently, the analysis of the wake development and the effect the TE devices have on the vortex shedding mechanism will be based on the Fine – DDES results. For completeness, the comparison between the URANS and DDES Fine mesh predictions is also discussed.


Figure 13 shows a comparison between the URANS and DDES on the Fine mesh for the plain airfoil case. Instantaneous vortical structures are visualised as isosurfaces of the Δ-criterion (Chong et al., 1990), coloured by streamwise vorticity. These are superimposed on isosurfaces of spanwise vorticity. The values for Δ and $\omega_z$ have been chosen so that the two surfaces overlap, where they both exist simultaneously. This method of visualisation was selected so that both the streamwise and the

spanwise vortex strength and complexity is visualised.

The artificial smoothing of the flow is apparent in the case of URANS. The spanwise vortices are only mildly three-dimensional and do not give rise to streamwise braids. In the URANS predictions, a typical von Kàrmàn vortex street is formed by opposite sign vortices arranged in an alternating fashion. However, these vortices have limited three-dimensionality and resemble wake

structures of significantly lower Re number (Bai and Alam, 2018). In the present case, the Reynolds number based on TE thickness is $Re_{TE} = 1.5 \times 10^5$, orders of magnitude higher than the critical Re number for bluff body wake flows



(Williamson, 1996), and significant three-dimensionality is expected. The vortices in the URANS case are also very quickly damped in the wake, despite the fact that the same 25M cells grid is used by both methods.

On the other hand, DDES results reveal the inherently strong three-dimensional character of the flow. According to DDES predictions, the spanwise vortices shed alternatively from the top and bottom trailing edges of the wing give rise to smaller streamwise hairpin vortices. These vortices play a central role in transfer of vorticity out of the core of the main vortex (Mittal and Balachandar, 1995) and appear as soon as the vortices are shed in the wake. The latter is not surprising given the high Re number and the turbulent boundary layer over the airfoil.


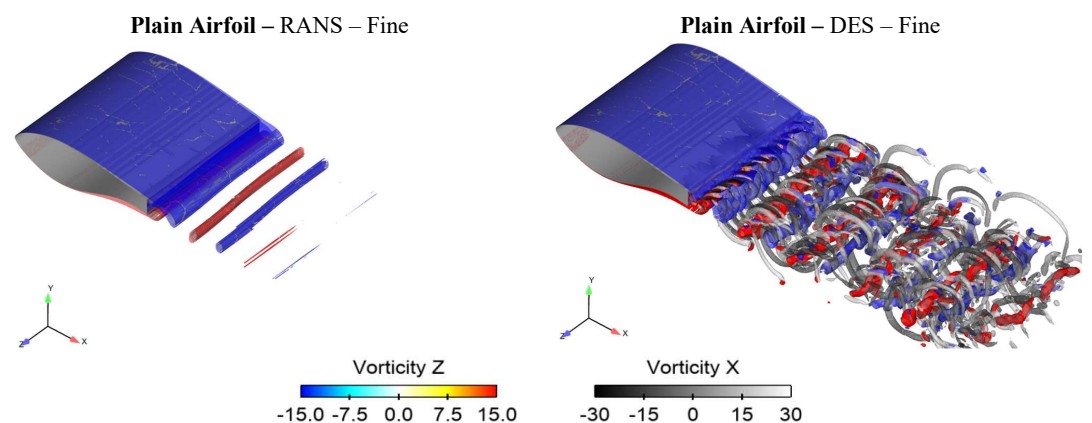

**Figure 13: Instantaneous isosurfaces of $\Delta = 10^5$ coloured by streamwise vorticity ($\omega_x$) and instantaneous isosurfaces of spanwise vorticity ($\omega_z = \pm 15$).**

Figure 14 offers the same instantaneous wake visualisation as Figure 13 for the cases with the different TE devices, as predicted by DDES on the Fine mesh. Significant differences are observed between the various cases. Perhaps the most complicated structures appear in the Splitter case, where oblique shedding is observed along with vortex dislocations (Tombazis and Bearman, 1997). Still, the wake remains smaller and less three-dimensional than the plain airfoil case (Figure 13, right). In the
Offset Cavity case, the streamwise braids appear less dense and maintain their regularity further downstream in the wake, compared to the Plain case. The spanwise vortices are also smaller in size and seem to break down sooner than in the uncontrolled case.

As discussed in more detail in Section 3.3, in the Flap case, the flap only affects the formation of the top vortex, while the lower vortex is shed uncontrolled. The wake hence appears more irregular than in the Offset Cavity case, but the spanwise von Kàrmàn vortices and the streamwise vortices are clearly identifiable. The wake is significantly more regular in the Flap +





Offset Cavity case, where the lower vortex formation is affected by the presence of the cavity plate. The spanwise vortices appear very well structured and the streamwise braids are not as pronounced.


Based on the analysis above it can be concluded that the TE devices all affect the formation of the von Kàrmàn vortices, which are present in all cases. All controlled cases have a smaller and less turbulent wake, in agreement with the previous section and with (Manolesos and Voutsinas, 2016). In general, the stronger the spanwise vortex, the stronger its three-dimensional character. Comparing the cases of Splitter and Offset Cavity, it appears that the distance of the control plate plays a significant

role in the vortex formation and strength. Finally, the comparison between the Flap and the Flap + Offset Cavity cases suggests that controlling only one side of the vortex street is not sufficient to create a well-structured and less turbulent wake.

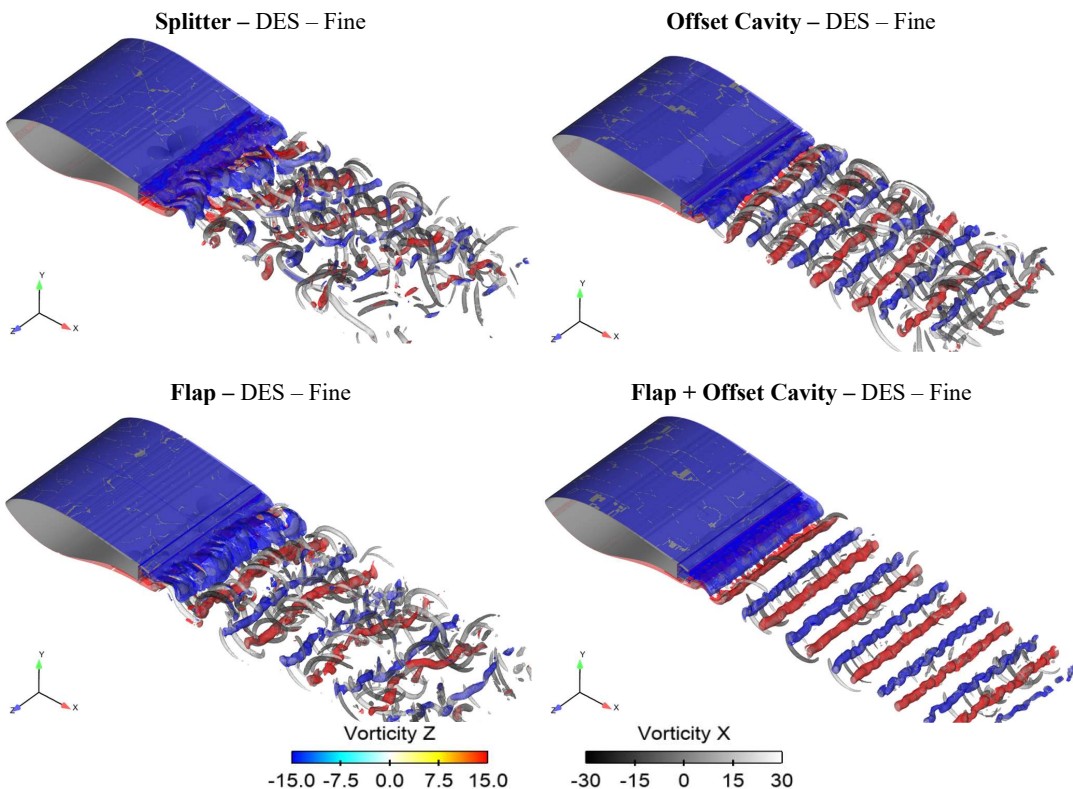

**Figure 14: Instantaneous isosurfaces of $\Delta = 10^5$ coloured by streamwise vorticity and instantaneous isosurfaces of spanwise vorticity ($\omega_z = \pm15$). DDES results on the Fine mesh.**



### 3.3 Vortex formation

In the following, the mechanics of vortex shedding from the blunt TE, with and without the TE devices, are described. This analysis is only concerned with the spanwise vortices and their shedding from the wing TE or the devices TE. Naturally, this is a highly three-dimensional flow and any description dealing only with spanwise concentrated vortices is bound to be incomplete. Still, the spanwise vortex shedding mechanism is the one that dominates the wake flow and is also the one that subsequently gives rise to the three-dimensional features. Further, it is the modification of this mechanism, i.e. the modification

of the roll up of the two shear layers into discrete vortices, that leads to the load changes for the airfoil and different St numbers in the wake. In this sense, it is considered of high importance to describe the flow in the airfoil wake. The description is based on the most accurate predictions, DDES on a Fine mesh, which are in good agreement with the experimental results.

Figures from Figure 15 to Figure 19 present instantaneous isosurfaces of the Q criterion (Hunt et al., 1988) of all the examined

cases. The shedding period (T) for each case has been split in six parts and seven snapshots are shown. The isosurfaces are not coloured by any other quantity to avoid confusion, given the complexity of the structures. Red arrows indicate the location of the main spanwise vortices, as interpreted from the analysis of the results. It is highlighted that the analysis was supported by flow animations (Papadakis and Manolesos, 2020), which cannot be included in the present document, but were fundamental in understanding the flow mechanisms. It is noted that in order to offer the reader a broader view of the results, a different

vortex identification criterion was selected in this case, compared to Figure 13 for example. The selection of the criterion does not affect the conclusions of the study.

Starting with the Plain airfoil case (Figure 15) and following the descriptions of (Gerrard, 1966; Tombazis and Bearman, 1997; Williamson, 1996) it is observed that the top vortex is fed with vorticity from the airfoil top boundary layer and grows in

size and strength. During this period the vortex remains attached to the top side of the blunt trailing edge until eventually it grows strong enough to draw the opposing shear across the wake. Then, opposite-sign vorticity is entrained between the top shear layer and the top vortex, which is when the feeding of vorticity cedes and the vortex is shed in the wake. In a similar manner, the lower spanwise vortex is fed with vorticity from the airfoil lower boundary layer and is cut off when it draws vorticity of opposite sign between itself and the feeding shear layer.


This 'feeding' mechanism is present in all cases, but the presence of the flow control devices alters when and where the vortex is shed in the wake. In the Splitter case, Figure 16, it appears that there is constantly a vortical structure at both edges of the blunt TE. In an alternating manner these structures grow and shed a vortex across the splitter plate. It is noted that when the vortices leave the wing TE, they are much smaller in size and strength compared to the Plain airfoil case. Furthermore, these

vortices are not immediately shed in the wake but grow further at each side of the Splitter plate, feeding from the adjacent shear layer (top or bottom), before they are eventually convected downstream. This mechanism has a higher frequency than



the Plain airfoil one (see also Figure 7) and explains the longer formation length and smaller wake thickness, observed in the experiments and simulations (see also Figure 9).

In the Offset Cavity case, Figure 17, the top vortex begins to form when the lower one is about to be shed into the wake. In that instant, they are both at the edges of the cavity plates. Then the top vortex grows, feeding from the top shear layer, while there is also a stationary vortical structure adjacent to the wing TE, between the top cavity plate and the wing TE. A similar structure exists at the bottom edge, between the wing and the lower plate. This structure is not stationary. As the top vortex grows, the lower structure is fed from the wing lower boundary layer and is elongated under the cavity lower plate. When the
top vortex grows strong enough to be shed in the wake, the lower elongated vortical structure is split into the lower vortex and a vortical structure remaining close to the wing lower TE. The lower vortex will shortly stand at the lower plate TE, before also being shed. The different behaviour of the vortical upper and lower structures it due to the different flow direction as it leaves the airfoil upper and lower side and it is directed into the wake and on the cavity plates. In the top side, the flow is directed towards the plate, while in the lower side it is directed away from it.

Figure 18 presents the Flap + Offset Cavity case. The mechanics are similar to the Offset Cavity case, with two notable differences. Firstly, the top vortex now forms in at the TE edge of the flap, connected to its boundary layer and, secondly, the lower elongated vortical structure is split after the top vortex is shed in the wake. The wake in general appears considerably better structured and regular, compared to all the other cases. The Flap changes the symmetrical wake pattern and reduces the
St number with regards to the Offset Cavity case.

Finally, Figure 19 shows the vortical structures for the Flap case. Here, the top vortex is located at the TE of the Flap, and begins to form when the lower vortex, already shed from the lower TE of the wing, passes below it. The top vortex then remains almost stationary, growing in strength and size before it is eventually shed in the wake. In the meantime, a new lower
vortex has been formed at the lower TE of the wing. The new vortex feeds from the wing lower boundary layer and is shed in the wake only after the top vortex has been convected. The lower vortex then travels under the flap, generating the top vortex as it passes below the flap TE. The wake is markedly more turbulent and less well-structured than the Flap + Offset Cavity case. It is conceivable that this is because the lower vortex is stronger and more three-dimensional than in the Flap + Offset Cavity case, where its formation was affected by the presence of the lower cavity plate.

**4 Conclusions**

A computational investigation of the flow past a flatback airfoil with and without TE control devices has been presented and the results have been compared with Wind Tunnel measurements. Both URANS and DDES modelling has been considered on two grids of different density, a Coarse with 5M cells and a Fine with 25M cells. Computations are compared with





measurements in terms of loads, wake frequencies and Stereo PIV measurements. Overall, if wake structure and frequencies
are of interest, then the highest fidelity simulation, DDES on a Fine mesh, should be preferred, as it provides the best agreement
with experimental data in all cases and parameters. If, on the other hand, only the effect of the TE devices on the loads is of
interest, then even the 2D URANS simulations can provide acceptable results. The simplest computational method, in
agreement with the experiments and all other numerical predictions, suggest that the Flap device outperforms all other
examined flow control solutions in terms of Lift and Lift to Drag ratio.


In more detail, 2D URANS predictions give a qualitatively correct estimation of the TE device relative performance. The load
variation for the Plain airfoil case, however, appears overestimated. There is fair comparison between the 3D URANS
simulations and the experimental data in terms of loads and flow quantities for the Plain airfoil case, especially on the Fine
mesh. For the cases with the TE devices, the 3D URANS loads predictions are in agreement with the measurements in terms
of mean values, however, vorticity shedding is under-predicted or completely absent from the simulations. This is evident in
the predicted St number being close to zero for the Flap, Cavity and Flap + Offset Cavity configurations. It is noted that
increasing the mesh size from 5M to 25M cells lead to similar results and no vorticity shedding.

On the contrary, DDES simulations are in better agreement with the experimental data for all cases.  The agreement is fair for
both the unsteady characteristics as well as the predicted loads. DDES modelling captures the unsteady character of the wake
on both grids. Loads, however, appear to be unsteady only on the fine grid case on most cases. Naturally, refining the grid
leads to a better comparison with the experimental data especially when comparing to Stereo PIV measurements and St
numbers.

Finally, using the most accurate predictions, DDES on the Fine mesh, the effect of the TE devices on the vortex formation and
shedding has been analysed. In the Plain airfoil case, the growth of the vortices as they receive vorticity from the airfoil
boundary layers is apparent, followed by the shedding of the vortices in the wake in an alternating manner. All TE devices
affect the shedding mechanism. In the Splitter case the vortices are formed on the airfoil upper and lower TE and they are
smaller in size compared to the Plain airfoil case when they are shed in the wake. In the Offset Cavity case, the top vortex
forms at the TE of the upper cavity plate, while the lower comes off an elongated vortical structure between the lower cavity
plate and the airfoil TE. In the Flap case the top vortex forms at the TE of the flap and is fed by its boundary layer. In the
Flap + Offset Cavity case, the wake is better structured and less three-dimensional than all other cases, with the top vortex
forming at the TE of the flap and the lower splitting off the lower elongated vortical structure, as in the Offset Cavity case.





**Acknowledgements**

Computational resources were provided by HPC Wales, which is gratefully acknowledged.

M Manolesos would like to acknowledge the contribution of the Supergen Early Career Researcher Fund Award by the Supergen Offshore Renewable Energy Hub, EPSRC.








**Figure 15: Snapshots of Q isosurfaces from the Plain airfoil case, DDES, Fine mesh results. Each period of the vortex shedding mechanism is split in six parts and a total of seven snapshots is shown. For each snapshot, two visualizations are shown. A 3D view of Q = 100 isosurfaces on the left and a side view of Q = 100 isosurfaces on the right. Superimposed red arrows indicate interpreted spanwise vortices.**

t/T = 0/6

The **upper vortex** is at the top Trailing Edge, while

the **lower vortex** is already part of the wake.

t/T = 1/6

The **upper vortex** grows remaining close to the wing, while

the **lower vortex** is convected downstream.

t/T = 2/6

The **upper vortex** has just been shed in the wake, when

a new **lower vortex** has formed.

t/T = 3/6

The **upper vortex** is convected downstream, while

the new **lower vortex** grows, remaining close to the wing.

t/T = 4/6

A new **upper vortex** is formed, when

the new **lower vortex** has just been shed.

The new **upper vortex** grows, remaining close to the wing, while

t/T = 5/6

the new **lower vortex** is convected downstream.

The new **upper vortex** grows further, while

t/T = 6/6

the **lower vortex** is convected further downstream.







**Figure 16:** Snapshots of Q isosurfaces from the Splitter case, DDES, Fine mesh results. Each period of the vortex shedding mechanism is split in six parts and a total of seven snapshots is shown. For each snapshot, two visualizations are shown. A 3D view of Q = 100 isosurfaces on the left and a side view of Q = 1.5 isosurfaces on the right. Superimposed red arrows indicate interpreted spanwise vortices.

t/T = 0/6

Vortical structures are always next to the wing top and bottom TE. The top vortex is convected downstream, while

The lower vortex travels downstream under the splitter.

The top vortex is convected downstream and the top vortical structure grows, while

t/T = 1/6

The lower vortex grows under the splitter.

A new top vortex is shed from the top vortical structure, while

The lower vortex is shed from the splitter.

t/T = 2/6

The lower vortex joins the wake.

A new top vortex travels above the splitter plate, while

t/T = 3/6

the lower vortical structure grows and the lower vortex is convected.

A new top vortex grows above the splitter plate, while

t/T = 4/6

the lower vortical structure grows and the lower vortex is convected further.

A new top vortex joins the wake, while

A new lower vortex is shed from the lower vortical structure.

t/T = 5/6

The top vortex is convected downstream and the top vortical structure grows, while

The lower vortex travels downstream under the splitter.

t/T = 6/6



Figure 17: Snapshots of Q isosurfaces from the Offset Cavity case, DDES, Fine mesh results. Each period of the vortex shedding mechanism is split in six parts and a total of seven snapshots is shown. For each snapshot, two visualizations are shown. A 3D view of Q = 100 isosurfaces on the left and a side view of Q = 100 isosurfaces on the right. Superimposed red arrows indicate interpreted spanwise vortices.





Figure 18: Snapshots of Q isosurfaces from the Flap + Offset case, DDES, Fine mesh results. Each period of the vortex shedding mechanism is split in six parts and a total of seven snapshots is shown. For each snapshot, two visualizations are shown. A 3D view of Q = 1.5 isosurfaces on the left and a side view of Q = 100 isosurfaces on the right. Superimposed red arrows indicate interpreted spanwise vortices.

**t/T = 0/6**

The **upper vortex** begins to form when

the **lower vortex** is beneath it. A **vortical structure** exists at the wing. The **upper vortex** remains close to the flap and grows in size, while

**t/T = 1/6**

the **lower vortex** joins the wake and the **vortical structure** is elongated. The **upper vortex** grows further, while

**t/T = 2/6**

the **lower vortex** is convected and the **vortical structure** is elongated further.

**t/T = 3/6**

The **upper vortex** is convected downstream, while

the **vortical structure** splits in two and a new **lower vortex** is shed.

**t/T = 4/6**

The **upper vortex** is convected further downstream, while

the **new lower vortex** passes under the lower plate. The **upper vortex** is convected downstream, while

**t/T = 5/6**

the **new lower vortex** grows remaining close to the plate Trailing Edge. A new **upper vortex** begins to form when

**t/T = 6/6**

the **lower vortex** is completely downstream of the lower plate.



**Figure 19.** Snapshots of Q isosurfaces from the Flap case, DDES, Fine mesh results. Each period of the vortex shedding mechanism is split in six parts and a total of 7 snapshots is shown. For each snapshot, two visualizations are shown. A 3D view of Q = 1.5 isosurfaces on the left and a side view of Q = 100 isosurfaces on the right. Superimposed red arrows indicate interpreted spanwise vortices.



## Appendix

Table 2: Mean and standard deviation (σ) values for lift and drag coefficients. Comparison between Experiments and DDES results on the Coarse and Fine mesh.

| | Experiment | | DDES Coarse | | | | DDES Fine | | | |
|---|---|---|---|---|---|---|---|---|---|---|
| | **Cl** | **Cd** | **Cl** | | **Cd** | | **Cl** | | **Cd** | |
| | mean | mean | mean | σ | mean | σ | mean | σ | mean | σ |
| **Clean Airfoil** | 0.399 | 0.042 | 0.353 | 0.017 | 0.058 | 0.002 | 0.361 | 0.064 | 0.083 | 0.006 |
| **Splitter** | 0.376 | 0.020 | 0.348 | 0.000 | 0.028 | 0.000 | 0.337 | 0.035 | 0.015 | 0.002 |
| **Offset Cavity** | 0.242 | 0.021 | 0.205 | 0.037 | 0.033 | 0.001 | 0.228 | 0.040 | 0.034 | 0.003 |
| **Flap** | 0.439 | 0.021 | 0.414 | 0.001 | 0.029 | 0.000 | 0.409 | 0.038 | 0.010 | 0.002 |
| **Flap + Offset Cavity** | 0.322 | 0.019 | 0.292 | 0.000 | 0.026 | 0.000 | 0.374 | 0.035 | 0.025 | 0.003 |


Table 3: Mean and standard deviation (σ) values for lift and drag coefficients. Comparison between 2D URANS and URANS on the Coarse and Fine mesh.

| | URANS 2D | | | | URANS Coarse | | | | URANS Fine | | | |
|---|---|---|---|---|---|---|---|---|---|---|---|---|
| | **Cl** | | **Cd** | | **Cl** | | **Cd** | | **Cl** | | **Cd** | |
| | mean | σ | mean | σ | mean | σ | mean | σ | mean | σ | mean | σ |
| **Clean Airfoil** | 0.283 | 0.018 | 0.095 | 0.073 | 0.347 | 0.052 | 0.016 | 0.361 | 0.366 | 0.067 | 0.060 | 0.015 |
| **Splitter** | 0.352 | 0.001 | 0.031 | 0.000 | 0.349 | 0.000 | 0.031 | 0.000 | 0.353 | 0.001 | 0.031 | 0.000 |
| **Offset Cavity** | 0.222 | 0.000 | 0.030 | 0.000 | 0.213 | 0.000 | 0.029 | 0.000 | 0.223 | 0.000 | 0.030 | 0.000 |
| **Flap** | 0.419 | 0.000 | 0.033 | 0.000 | 0.419 | 0.000 | 0.034 | 0.000 | 0.419 | 0.001 | 0.033 | 0.000 |
| **Flap + Offset Cavity** | 0.308 | 0.001 | 0.028 | 0.000 | 0.285 | 0.000 | 0.028 | 0.000 | 0.307 | 0.000 | 0.028 | 0.000 |




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



## 5 Code availability

Code available upon request

## 6 Data availability

All data available upon request.

## 7 Video supplement

Animations. https://doi.org/10.5281/ZENODO.3662124

## 8 Supplement link (will be included by Copernicus)

## 9 Team list

Dr George Papadakis

School of Naval Architecture & Marine Engineering, National Technical University of Athens, Athens, Greece

Dr Marinos Manolesos

College of Engineering, Swansea University, Bay Campus, Fabian Way, SA1 8EN, Swansea, UK

## 10      Author contribution

Dr George Papadakis performed the numerical calculations. For the analysis and investigations of the numerical and experimental results the contribution of the authors was equal. Both authors contributed equally to manuscript authorship.