# Peer review of "The flow past a flatback airfoil with flow control devices: Benchmarking numerical simulations against wind tunnel data"

_Wind Energy Science, 2020_

## Referee Comment (RC1) · Anonymous Referee #1 · 24 Feb 2020

Manuscript focuses on the CFD-based analysis of a flatback airfoil with various trailing edge treatments and comparison of the numerically obtained results using RANS, URANS, and DES with wind-tunnel measured results. The manuscript addresses the important topic of design and analysis of unconventional airfoils for wind turbine rotors. It is well written although the ratio of number of illustrations to number of lines of text is high and this negatively impacts the readability of the manuscript.

Specific comments:

One issue is the focus on 2D airfoils where it is important to note that on 3D blades, the flow unsteadiness and particularly the Von Karman vortex shedding encountered

in 2D may well be mitigated by spanwise pressure gradients and geometry changes. This brings up the question if the trailing edge treatment studied in this manuscript are effective and/or are needed on a wind turbine blade? Especially in the inboard region of the blade, where the flatback airfoils are being applied, spanwise pressure gradients and geometry changes are significant causing the flow to be very three dimensional.

Line 104. These trailing edge treatments may affect the high angle of attack characteristics including maximum lift coefficient and stall angle. By limiting the analysis to an angle of attack of zero degrees, the impact of these treatments on this important part of the operating envelope of airfoils is not assessed. Based on this, I would be careful recommending any of the trailing-edge treatments.

Line 188. Concern is that because of the deflection, the flap configuration is not constant in the spanwise direction and that this spanwise variation, not captured in the numerical simulations, causes the discrepancies between experiment and CFD.

Line 273. Delta-criterion is used for the streamwise vorticity. However, in the corresponding figures, Omega_x is listed. In Fig. 13, Delta is listed in caption but Omega_x in the figure. Consistently use Omega_x or, if this causes any issues, more clearly explain the Delta-criterion.

Technical corrections:

Line 118. 0.62 h. Should this be 0.62 h_TE?

Line 120. Forces non-dimensionalized using the chord of the baseline airfoil? Want to be precise because in line 118 the chord of the flap is mentioned.

Line 121. "misalignment of the model has been aligned of the model has been allowed for". Please reword.

Lines 137&138. Please reword.

Line 200. Fig. 6 is very unclear. As is this figure is less than useful.

---

## Referee Comment (RC2) · Anonymous Referee #2 · 23 Apr 2020

The primary purpose of the paper is to describe a computational study of the effects of various trailing edge passive flow control devices for a flatback airfoil, including an investigation of the physical processes involved. A secondary purpose is to validate several CFD turbulence modeling approaches for this flow by comparing results to experimental data. The paper is mostly successful at both, although some important details are neglected.

Specific Comments: 1. It is not clear how the trailing edge devices were included in the CFD model. Was the grid modified to wrap around the new geometry? Or is there an immersed boundary technique applied? The authors should strongly consider showing

at least one example of a mesh for one of the device configurations.

2. IDDES is named as the hybrid RANS/LES turbulence model, but then this is abbreviated to DDES. This creates confusion as to which model was actually used and how it was applied. IDDES is capable of modeling the outer part of attached turbulent boundary layers in LES mode, while in DDES the attached boundary layers are always modeled in RANS mode. IDDES and DES have different model equations. Please clearly state which model implementation was used and, if IDDES, whether the attached TBL regions were quasi-steady RANS regions or LES (I strongly suspect the former, given the stated grid resolution).

3. At Re_c=1.5e6, one might expect boundary layer transition to play a key role in predictions of lift and drag, and possibly the wake region. How was transition handled?

4. It would be very illuminating to perform at least one simulation at the experimental aspect ratio, to study any end effects, if present. Absent this, quantifying the span-wise correlation length of velocity fluctuations in the wake would give confidence that the span-wise extent of the domain is long enough to at least approximate the large-aspect ratio case. Another way to explore this issue would be to see if spanwise periodic BC's give different results?

5. I had difficulty reconciling the high experimental wake fluctuation amplitude with the modest experimental Reynolds stress field for the flap-only case.

Technical Corrections: 1. Sentence on "misalignment" on the bottom of page four is unclear. 2. Are there any experimental measurement uncertaintites available to improve the validation exercise? 3. The term "loads" is used to describe mean aerodynamic loads, which may make sense to the wind energy practitioner. However, loads can also be unsteady so consider using the term "mean loads".

---

## Author Comment (AC1) · 9 May 2020

We would like to sincerely thank the reviewer for their constructive review and valuable comments which helped us provide an improved presentation of our work. Please find below answers to the reviewer's comments .

**General Comment Manuscript focuses on the CFD-based analysis of a flatback airfoil with various trailing edge treatments and comparison of the numerically obtained results using RANS,URANS, and DES with wind-tunnel measured results. The manuscript addresses the important topic of design and analysis of unconventional airfoils for wind turbine rotors. It is well written although the ratio of number of illustrations to number of lines of text is high and this negatively impacts the readability of the manuscript.**

Thank you for the kind comment, regarding figures, we believe all illustrations are useful and serve their purpose in the manuscript. The vortex dynamics discussion figures have been given independently in an Appendix. The readability of the manuscript should be improved when the paper adopts the journal format.

**Comment 1. One issue is the focus on 2D airfoils where it is important to note that on 3D blades, the flow unsteadiness and particularly the Von Karman vortex shedding encountered in 2D may well be mitigated by spanwise pressure gradients and geometry changes. This brings up the question if the trailing edge treatment studied in this manuscript are effective and/or are needed on a wind turbine blade? Especially in the inboard region of the blade, where the flatback airfoils are being applied, spanwise pressure gradients and geometry changes are significant causing the flow to be very three dimensional.**

The reviewer is correct. The following sentence has been added in the revised manuscript to highlight the limitations of this study.
*"It is noted that the study is limited to extruded airfoil with no twist or profile change and extension of any findings to 3D rotating blades would require further validation. "*

**Comment 2 Line 104. These trailing edge treatments may affect the high angle of attack characteristics including maximum lift coefficient and stall angle. By limiting the analysis to an angle of attack of zero degrees, the impact of these treatments on this important part of the operating envelope of airfoils is not assessed. Based on this, I would be careful recommending any of the trailing-edge treatments.**

The reviewer rightly mentions the very limited AoA range of this study. Indeed, the higher AoA are of significant interest both for the numerical approach and for the actual results and analysis. However, for this work we decided to limit the scope to a single AoA so that there is a focus (a) on the different numerical schemes and (b) on the analysis of the wake structures. Indeed, even at 0deg, the 3D unsteady bluff body wake flow is challenging for CFD methods and rich in flow dynamics.

Preliminary simulations at high AoA, beyond Cl_max, show that the von Karman-like wake is mixed with Stall Cell like structures, increasing the complexity and level of difficulty well beyond the scope of the present submission. We believe the single AoA data are sufficient to achieve the objectives of this study, which are (a) to examine which numerical approach is most suitable to study the flow in question and (b) to provide insight into the effect of the various flow control devices on the airfoil wake.

**Comment 3. Line 188. Concern is that because of the deflection, the flap configuration is not constant in the spanwise direction and that this spanwise variation, not captured in the numerical simulations, causes the discrepancies between experiment and CFD.**

The reviewer is correct. This is exactly why we mentioned it. Given the good agreement between the predictions and the experiments, it is concluded that the effect of this deflection does alter the findings of this study.

**Comment 4 Line 273. Delta-criterion is used for the streamwise vorticity. However, in the corresponding figures, Omega_x is listed. In Fig. 13, Delta is listed in caption but Omega_x in the figure. Consistently use Omega_x or, if this causes any issues, more clearly explain the Delta-criterion.**

Omega_x stands for streamwise vorticity. The Δ isosurfaces are coloured with streamwise vorticity. In Fig 13 and 14 two types of isosurfaces are overlaid, Δ and spanwise vorticity. The former are coloured by streamwise vorticity and the latter by spanwise vorticity. This is why both are mentioned and shown in the legend. The word *overlaid* has been added to the captions to clarify this.

**Technical Comments**

**1.** Line 118. 0.62 h. Should this be 0.62 h_TE?

Yes, this has been corrected in the manuscript.

**2.** Line 120. Forces non-dimensionalized using the chord of the baseline airfoil? Want to be precise because in line 118 the chord of the flap is mentioned.
This has been clarified and the sentence below has been added to the manuscript:

*"In the remaining of this article all quantities are non-dimensionalized using the baseline airfoil chord and the free stream velocity, as reference values, unless otherwise stated. "*
**3.** Line 121. "misalignment of the model has been aligned of the model has been allowed for". Please reword.

This sentence now reads:

*"A constant misalignment of the model has been allowed for in the results. "*

**4.** Lines 137&138. Please reword.

This has been corrected. The sentence now reads:
*"The error bars are based on the standard deviation values. For completeness the relevant values are also given in Table 2 and Table 3, in the Appendix. "*

**5.** Line 200. Fig. 6 is very unclear. As is this figure is less than useful.

The figure has been replaced, given below for convenience

[Figure]

**Figure 1: Normalized Power Spectral Density of the velocity time series for the Plain airfoil and all the TE device cases from Fine IDDES simulations. The vertical velocity is considered at (x, y) = (1.57c, 0) see also Figure 2. Strouhal number is defined based on the trailing edge thickness, $St = f h_{TE}/V_\infty$**

---

## Author Comment (AC2) · 9 May 2020

We would like to sincerely thank the reviewer for their constructive review and valuable comments which helped us provide an improved version of our work. Please find below answers to the reviewer's comments.

**Comment 1. It is not clear how the trailing edge devices were included in theCFD model. Was the grid modified to wrap around the new geometry? Or is there an immersed boundary technique applied? The authors should strongly consider showing at least one example of a mesh for one of the device configurations.**

The grid was modified to wrap around the TE device. A relevant figure (Figure 1) has been added to show this.

**Comment 2.IDDES is named as the hybrid RANS/LES turbulence model, but then this is abbreviated to DDES. This creates confusion as to which model was actually used and how it was applied. IDDES is capable of modeling the outer part of attached turbulent boundary layers in LES mode, while in DDES the attached boundary layers are always modeled in RANS mode. IDDES and DES have different model equations. Please clearly state which model implementation was used and, if IDDES, whether the attached TBL regions were quasi-steady RANS regions or LES (I strongly suspect the former, given the stated grid resolution).**

Changed DDES to IDDES in the text and graphs. Indeed, as the reviewer states the boundary layer regions were RANS regions. This has been added to the text.

*"The IDDES model considers a RANS zone in the boundary layer region and switches to LES in the wake."*

**Comment3. At Re_c=1.5e6, one might expect boundary layer transition to play a key role in predictions of lift and drag, and possibly the wake region. How was transition handled?**

All experimental data are free transition data. Only free transition data are available from the device cases from the experimental study. For the plain airfoil the effect of fixing transition was minimal, as drag is dominated by base drag and maximum lift angle does not change. The reason for this is the reduced adverse pressure gradient of the flatback airfoil.

In order to clarify this, for the experimental case the following sentence has been added to the revised manuscript.

*"Only free transition experimental results were available for these cases. "*

Regarding the numerical predictions the flow was considered fully turbulent to exclude transition model uncertainties from the comparison (especially if the transition point fluctuates due to the unsteadiness of the flow). However, we plan to add transition modelling in future work. This sentence has been added to the text:

*"All the simulations consider the flow fully turbulent to exclude possible uncertainties related to transition modelling."*

**Comment 4. It would be very illuminating to perform at least one simulation at the experimental aspect ratio, to study any end effects, if present. Absent this, quantifying the span-wise correlation length of velocity fluctuations in the wake would give confidence that the span-wise extent of the domain is long enough to at least approximate the large-aspect ratio case. Another way to explore this issue would be to see if spanwise periodic BC's give different results?**

The reviewer correctly states that quantifying the correlation length of the velocity fluctuations is important. To that end we added a figure showing the Pearson correlation coefficient for the Cl time signals and added a relevant paragraph. We chose the Cl signal instead of the wake fluctuations due to the limited spanwise points in the wake that the velocity was recorded. The text and figure are given below for convenience.

*In order to quantify the spanwise correlation of the flow for each device, the Pearson correlation coefficient (r) of the Cl time signal at different spanwise locations is presented in Figure 15. Values of 1, -1 indicate strong correlation between the signals (positive and negative, respectively) while value of 0 suggests no correlation. The correlation coefficient is calculated with respect to the midsection (located at $z/b = 0$ in Figure 15). It is evident that the TE devices significantly alter the spanwise correlation of the flow. When the plain configuration is examined a correlation length of $\lambda = 0.5b$, where b is the wing span, or $\lambda = 5h_{TE}$ is identified, in agreement with (Metzinger et al., 2018). For the flap case, the correlation length remains large with $\lambda = 5.9h_{TE}$. It is noted that in the flap case the lower vortex is shed uncontrolled and this could explain the strong coherence in the wake. The Splitter and Offset Cavity cases have the correlation length drops to $\lambda = 2.5h_{TE}$ and $\lambda = 2.7h_{TE}$, respectively. The weakest spanwise coherence is observed for the Flap + Offset Cavity with $\lambda = 0.7h_{TE}$. The preceding analysis is also in agreement the isosurfaces shown in Figure 14. Indeed, as the figure suggests, the spanwise correlation length of the vorticity isosurfaces in the flap and offset cavity configuration is much smaller when compared to the flap one. Finally, it is noted here that using the wake velocity fluctuations for the preceding analysis yields similar results, however, the Cl was employed since it was already available at all spanwise stations.*

[Figure]

*Figure 1. Pearson correlation coefficient of the lift coefficient (Cl) time-series with respect with the midsection (placed at z/b=0) for the various configurations. High positive values indicate strong positive correlation and highly negative values suggest a strong negative correlation.*

**Comment 5. I had difficulty reconciling the high experimental wake fluctuation amplitude with the modest experimental Reynolds stress field for the flap-only case**

The experimental data were analyzed in the reference given at the end of the reply to this comment. Figure 22 from that reference (also given below for convenience) shows that while the peak amplitude for the dominant frequency for the Flap case is higher than the plain airfoil for example, the amplitude of all other frequencies are lower. This would justify a high experimental wake fluctuation amplitude (shown in fig. 7, right, of our revised submission) and the modest Re stress field (shown in fig. 11 of the revised submission). In other words, fig. 7 only concerns the dominant frequency, while the Re stress contour contains information from all frequencies.

[Figure]

*Figure 22. Frequency spectrum from the hot wire measurements for the examined TE devices and the plane airfoil.*

Since this discussion is focused only on the experimental side of this investigation it was decided not to include it in the present submission.

Manolesos, M., Voutsinas, S.G., 2016. Experimental Study of Drag-Reduction Devices on a Flatback Airfoil. *AIAA J.* 54, 3382–3396. https://doi.org/10.2514/1.J054901

**Technical Comments:**
**1.** Sentence on "misalignment" on the bottom of page four is unclear.
This has been corrected. The sentence now reads:
*"A constant misalignment of the model has been allowed for in the results."*

**2.** Are there any experimental measurement uncertaintites available to improve the validation exercise?

The following paragraph has been added to the manuscript.

*"A detailed uncertainty analysis can be found in (Manolesos and Voutsinas, 2016), but a short overview is given here for completeness. The 95% confidence interval for the lift and drag values are 1% and 4%, respectively. For the hot wire frequency spectrum, the frequency step was 1.95Hz, while for the Stereo PIV measurements, the minimum resolvable velocity is 1.5% . Any velocities lower than this should not be trusted. "*

**3.** The term "loads" is used to describe mean aerodynamic loads, which may make sense to the wind energy practitioner. However, loads can also be unsteady so consider using the term "mean loads".

The term 'loads' has been replaced with 'forces'.